# Depth dependent stress revealed by aftershocks

Peter Shebalin[1] & Clément Narteau [2]

Characterising the state of stress in the brittle upper-crust is essential in mechanics of faulting, industrial production processes, and operational earthquake forecasting. Nevertheless, unresolved questions concern the variation of pore-fluid with depth and the absolute strength on tectonically active faults. Here we show that, along the San Andreas fault system, the time-delay before the onset of the power-law aftershock decay rate (the c-value) varies by three orders of magnitude in the first 20 km below the surface. Despite the influence of the lithostatic stress, there is no continuous change in c-value with depth. Instead, two decay phases are separated by an abrupt increase at an intermediate depth range of 2–5 km. This transitional regime is the only one observed in fluid-injection-induced seismic areas. This provides strong evidence for the role of fluid and a porosity reduction mechanism at depth of few kilometres in active fault zones. Aftershock statistics can then be used to predict changes in differential shear stress with depth until the brittle-ductile transition is reached.

[1] Institute of Earthquake Prediction Theory and Mathematical Geophysics, 84/32 Profsouznaya, Moscow 117997, Russia. [2] Institut de Physique du Globe de Paris, Sorbonne Paris Cité, Univ Paris Diderot, UMR 7154 CNRS, 1 rue Jussieu, 75238 Paris Cedex 05, France. Correspondence and requests for materials should be addressed to C.N. (email: narteau@ipgp.fr)

Despite an increasing number of observations along the San Andreas fault system, no consensus has been reached about the absolute strength of the major strike-slip faults that accommodate the right-lateral motion between the Pacific and North American plates[1–7]. The absence of heat flow anomaly near the major fault segments[4] as well as the apparent high angle between these fault planes and the direction of the maximum horizontal compressive stress[1] suggest an extremely weak fault in term of friction coefficient[8]. Other arguments related to stress rotation in transpressional plate boundaries support a strong-fault hypothesis[6]. In all debates, the role of fluids has to be considered because pore pressure may linearly compensate for the overburden pressure at depth[5,9]. Under such condition, the frictional strength of faults decreases significantly as the pore pressure increases in excess to hydrostatic[10]. Hence, changes in pore pressure have been proposed as a generic mechanism for seismicity and aftershock sequences[11–13]. This mechanism has been extensively studied in mining or fluid injection sites where fluid pressure and stress diffusion may trigger seismic events or considerably change the earthquake rate[14–17]. However, there is still no well-established relation between earthquake statistics and pore pressure along active tectonic faults.

Direct measurements of the stress magnitude in the brittle upper-crust relies on drilling methods using hydraulic fractures and wellbore breakouts[1,18]. Due to the needs for large and homogeneous samples to examine the level of stress along active fault zones, indirect measurements are still required and earthquakes continue to provide the most informative data for comparative analysis[19–21]. Stress directions are generally derived from inversion of focal mechanisms[22,23]. Independently, earthquake catalogues are increasingly used to infer changes in stress intensity from deviations of well-known distributions in statistical seismology[24]. The two main parameters under investigation are the slope of the earthquake size distribution, the $b$-value[25], and the time delay before the onset of the power-law aftershock decay rate, the $c$-value[26]. As the time for nucleation and growth of brittle cracks is likely to decrease with the level of stress[27], it was proposed that $b$ and $c$-values reflect the same failure mechanisms within the process zone and the aftershock area during and after the dynamic rupture, respectively. Then, as shown from observations, they both exhibit the same negative dependence on stress[24,25].

Here, we concentrate on aftershock sequences along the San Andreas fault system to evaluate how the $c$-value varies with depth. We show that a sharp increase in $c$-value at intermediate depth of 2–5 km separates two decay phases with different slopes. These three different phases of the $c$-value vertical profile are consistent with variations in stress predicted by the Anderson's faulting theory and the Coulomb faulting theory considering a porosity reduction mechanism at intermediate depth. We conclude that such variations of stress magnitude can naturally explain changes in earthquake statistics with depth, as well as the apparent weakness of faults along major strike-slip faults in California.

## Results

**Vertical $c$-value profiles in California**. We examine aftershock sequences along the San Andreas fault system using the hypocenter information of the modern waveform relocated catalogues for southern and northern California from 1984 to 2016[28–30]. To identify mainshocks, we deselect earthquakes of magnitude smaller than $M$ which are within a $0.02 \times 10^{0.5M}$ km radius circle during the first $0.04 \times 10^{0.55M}$ days after a magnitude $M$ event[31]. Keeping the same spatial scaling, earthquakes that precede larger events by less than 24 h are classified as potential foreshocks. All

the remaining events are mainshocks. Their respective aftershocks are selected in the time interval $[t_{\text{start}}, t_{\text{stop}}]$ after the mainshock time using again the same spatial scaling. For $t_{\text{start}} = 10$ s, $t_{\text{stop}} = 1$ day and $M \geq 2.5$, we obtain non-overlapping aftershock sequences from which we can analyse the early aftershock decay rate.

Another critical issue of our declustering method is that the selected earthquakes are classified in ranges of magnitude. Here, we focus only on $2.5 < M^{\text{M}} < 3.5$ mainshocks and $1.8 < M^{\text{A}} < 2.8$ aftershocks. By analysing only large aftershocks of small mainshocks, we do not only reduce artefacts related to catalogue completeness over short time, we also study narrow ranges of magnitude over which $c$ keeps a relatively large and constant value[32,33]. To reduce the impact of faulting style on $c$-values, we select only mainshocks along a set of well-defined subvertical strike-slip faults in narrow bands of 15 km wide down to focal depths of 20 km. Ultimately, we analyse 3000 aftershock sequences comprising 4839 events within the first day after the mainshock (1668 and 3295 in southern California, 1332 and 1544 in northern California). Then, we stack aftershocks according to the mainshock time to compensate for the small number number of events in each sequence.

We use uniform priors in a Bayesian estimation of the parameters of the Modified Omori Law,

$$\Lambda(t) = \frac{K}{(c+t)^p},\qquad(1)$$

to characterise the exponent $p$ of the power-law aftershock decay rate and the duration $c$ of the early stage of aftershock activity that does not fit with this power-law regime[34]. From the entire stacked sequence, Fig. 1 shows the Bayesian posterior densities of $\{c, p\}$ for shallow (4–6 km) and deep (9–13 km) aftershocks. These two distributions do not overlap below their 99% interior range, so that there are statistically significant differences between shallow and deep aftershock sequences along strike-slip faults in California. The incompatibility between these two distributions resides more in the $c$-value than in the $p$-value as indicated by the marginal posterior distributions of these two parameters (Fig. 1). This suggests that the time delay $c$ is more sensitive to depth than the power-law exponent $p$ (see "Methods" section). Hence, we

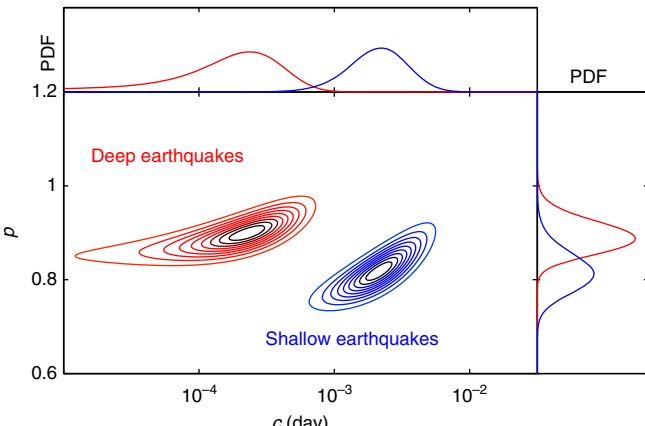

**Fig. 1** Influence of depth on the aftershock decay rate in zones of strike-slip faulting in California. Bayesian posterior densities of $\{c, p\}$ in two depth intervals of 4–6 km (blue, 557 events) and 9–13 km (red, 556 events) considering a single stack of aftershocks for all the major strike-slip faults. Contours are the deciles of each distribution. Curves show the probability distribution functions of the marginal posterior distributions of $p$ (left) and $c$ (top)

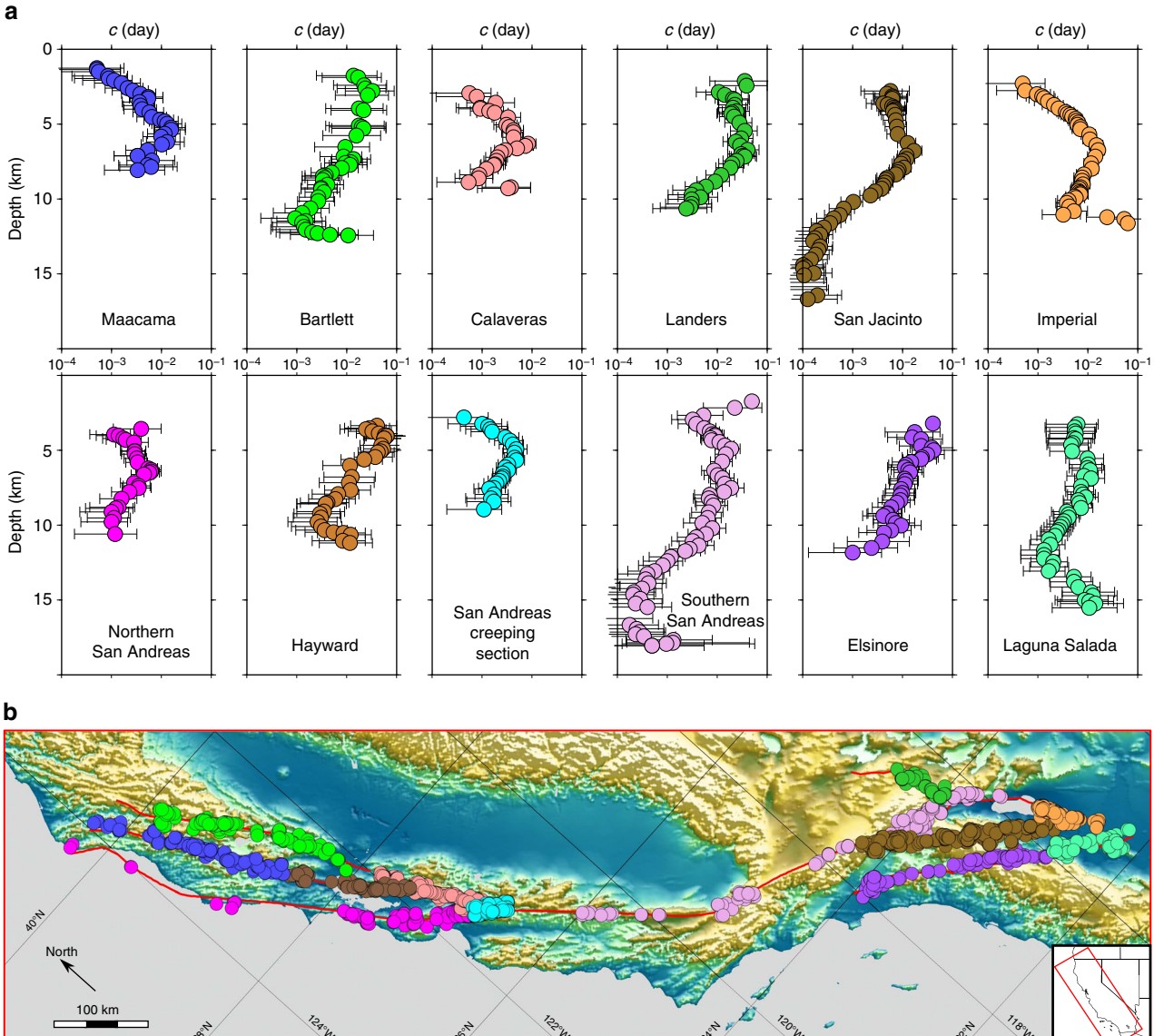

**Fig. 2** Influence of depth on the time delay before the onset of the power-law aftershock decay rate along major strike-slip faults in California. **a** Logarithm of the most probable $c$-value for $p = 1$ with respect to the mean depth of the selected aftershocks within the 3 km depth bins along individual faults. Error bars show the 95% Bayesian credibility regions for the $c$-value. **b** Major strike-slip faults in California and location of the selected mainshocks (colour code as in **a**)

document only variations in $c$-value estimating the parameters of the original Omori formula $\Lambda(t) = K/(c + t)$. Thus, we reduce the uncertainty imposed by the positive dependence on the estimation of both $\{c, p\}$-values[34]. In addition, we can also recover the variation of $c$-value using the $\langle t_g \rangle$-value, the geometric mean of aftershock times over the same time period $[t_{start}, t_{stop}]$ (see "Methods" section). This non-parametric estimator of the time delay before the onset of the power-law aftershock decay rate is free from uncertainties related to the fitting procedure and do not rely on the validity of the Modified Omori Law[35].

For each strike-slip fault under investigation, we isolate the corresponding stack of aftershocks and select them according to depth using a sliding window with a thickness of 3 km and a step of 0.3 km. Figure 2 shows the evolution of $c$-values with respect to depth for all these faults. Despite some variations related to the spatial distribution of seismicity in these specific areas and overlapping error bounds, a similar behaviour may be observed across the entire California: a sharp increase in $c$-value at shallow

depth ($\leq 5$ km) is followed by a continuous decrease to depth of about 15 km. As a result, the $c$-value may drop by more than two orders of magnitude from 15 mn at depth of 5 km to 20 s in the neighbourhood of the brittle-ductile transition zone. These estimates are obtained from independent sets of events, mainly in southern California where a larger number events occur over a wider depth range (e.g., San Andreas and San Jacinto faults). In all places, the continuous change in $c$-value is due to the moving step of the overlapping sliding-window for event selection, which gradually replaces a small proportion of aftershocks in the stacks. Finally, the dependence of the $c$-value on depth is tested by checking that similar signals can be recovered using the $\langle t_g \rangle$-value and a different declustering procedure[36–38] (see "Methods" section).

To explain these vertical $c$-value profiles, the stress magnitude at depth may be directly estimated from the Anderson faulting theory and limited by Coulomb faulting theory, which gives the maximum values of the differential shear stress before failure

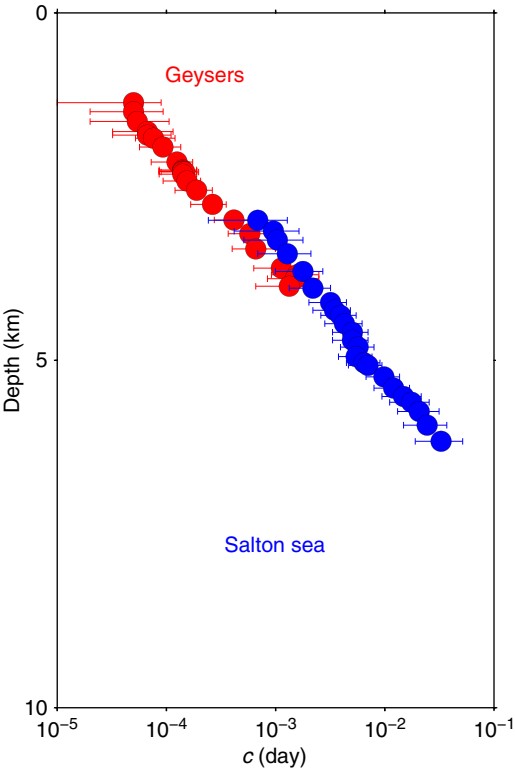

**Fig. 3** Influence of depth on the time delay before the onset of the power-law aftershock decay rate in zones of fluid-injection-induced seismicity. Logarithm of the most probable $c$-value for $p = 1$ with respect to the mean depth of the selected aftershocks within the 3 km depth bins in the Geysers (red) and Salton Sea (blue) areas. Error bars show the 95% Bayesian credibility regions for the $c$-value

(see "Methods" section). Considering a constant friction coefficient[8], the increase of the lithostatic stress with depth results in a linear increase of the minimum differential stress to initiate slip along pre-existing faults. Then, the systematic decrease in $c$-value in the depth interval from 5 to 15 km agrees with previous studies concluding that the time delay before the onset of the aftershock decay rate is negatively correlated to the differential shear stress[24,32,39,40]. On the opposite, the abrupt increase in $c$-value occurring in the shallow part of the crust (<5 km) cannot be explained by the linear increase of the lithostatic stress. To investigate this anomalous behaviour, we analyse the triggered seismicity in the neighbourhood of two industrial injection sites[41,42], the Geyser and the Salton Sea geothermal plants. Using all earthquakes in two circles of 10 km radius centred at (38°06″ N, 122°48″W) and at (33°12″N, 115°36″W), we apply the same procedure as along strike-slip faults to identify mainshock-aftershocks sequences. Figure 3 shows the evolution of the $c$-value with respect to depth for these two geothermal areas. In Geysers, a zone where all the seismicity is triggered by fluid injection, there is no earthquake below 5 km. In Salton Sea, where natural and fluid-induced earthquakes coexist, seismic events are observed to depth of 8 km. Despite all the differences between these two sites, they both exhibit the same $c$-value behaviour with respect to depth: a drastic increase of more than three orders of magnitudes from depth of 2 km to depth of 6 km. Such a behaviour can be directly compared to $c$-value variations at shallow depth along strike-slip faults, thereby shedding light on the role of fluid in these active fault zones.

In geothermal areas, the observed seismicity and local stresses are directly correlated to fluid-injection operations. The triggered seismic activity may then be associated with a relaxation process of stress and pore pressure in both space and time. Such a mechanism may naturally explain the increase in $c$-value with depth, i.e., with the distance from the injection source. Along strike-slip faults, we infer that a porosity reduction mechanism at depth of few kilometres could play the same role and strongly reduce the strength of pre-existing fractures.

In order to account for a porosity reduction mechanism in the Coulomb faulting theory, we propose a model for the evolution of the pore pressure with depth (Fig. 4a, see also "Methods" section). Since we concentrate on statistical properties that are averaged over space and time, we consider that pore pressure does not operate a sharp jump from its hydrostatic to its near-lithostatic regimes. Instead, below a threshold depth $z_c$, we assume that the transition between these two regimes takes the form of an exponential relaxation with a characteristic thickness $L$ (Fig. 4a). Combining the fault strength derived from this pore pressure profile with a creep rheology usually used to describe the brittle-ductile transition at larger depth (see "Methods" section), Fig. 4b shows the strength envelop for a friction coefficient $\mu = 0.75$. The variation of $c$-value is superimposed on this strength envelop using a different axis on the same graph. The comparison between these two signals reveal striking similarities at all depths. At intermediate depth between 2 and 5 km (i.e., $z_c = 2$ km and $L = 1$ km), the transition from hydrostatic to near-lithostatic pore-pressure conditions gives the depth-range within which the $c$-value increases. Above and below this transition layer, the rapid and slow increases in $c$-value are governed by the increase in lithostatic stress under hydrostatic and near-lithostatic pore-pressure conditions, respectively. Finally, for depth larger than 15 km, intercrystalline plasticity is thermally activated, creep dominates and the differential shear stress collapses. A transition which is also captured by the $c$-value.

## Discussion

The depth-dependent behaviour of the $c$-value is not a consequence of background contamination or an artefact related to the working assumption of $p = 1$. It can be independently recovered for southern and northern California estimating simultaneously the background seismicity and both the $c$ and $p$-values (see "Methods" section). However, local properties along faults in California are likely to differ and, because we rely on a limited amount of data, they cannot be individually inferred from aftershocks. This is why the overall effect of the confining stress and pore-pressure should be averaged at the length scale of California considering only events along major strike-slip faults.

The vertical $c$-value profiles presented in Figs. 2 and 4 illustrate how the differential stress control the duration of the preliminary phase of aftershock sequences at all depths across the brittle upper-crust. Similar results have been obtained for the slope $b$ of the earthquake size distribution[43]. A monotonic decrease in $b$-value between 5 and 15 km as well as a rapid increase below the brittle-ductile transition may be directly linked with similar behaviours of the $c$-value over the same depth intervals. In addition, an increase in $b$-value with respect to depth has also been documented at shallow depth (<5 km), but not interpreted. This symptomatic behaviour can reflect the porosity reduction mechanism revealed by aftershocks (Fig. 4). In this case, the magnitude of stress usually proposed from the slope $b$ of earthquake size distribution[44] may be overestimated and warrant further investigation.

Using small magnitude events occurring along strike-slip faults in California, aftershock statistics show that the characterisation of the state of stress in the brittle-crust should account for near lithostatic pore-pressure conditions. These conditions appear to

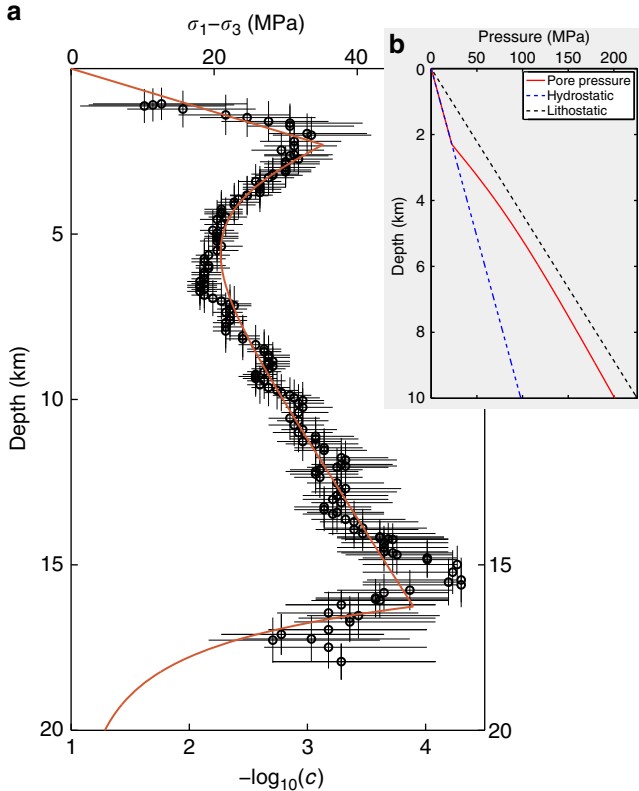

**Fig. 4** Differential shear stress and the time delay before the onset of the power-law aftershock decay rate with respect to depth in California. **a** Logarithm of the most probable $c$-value for $p = 1$ with respect to the mean depth of the selected aftershocks within the 3 km depth bins (see all mainshocks in Fig. 2b). Error bars show the 95% Bayesian credibility regions for the $c$-value. The differential shear stress (red line) is derived from the Mohr–Coulomb theory using a coefficient of friction $\mu = 0.75$, a porosity reduction mechanism at a depth of 2 km and a plastic flow law for wet quartzite (see "Methods" section and Table 1). **b** Evolution of pore (red), lithostatic (black), and hydrostatic pressures (blue) with respect to depth

be associated with a pore-reduction mechanism as a result of a loss of structural connectivity and fluid trapping within the fracture networks of active fault zones. As predicted by theory and observed from aftershocks, this most likely occurs in the vicinity of a percolation threshold at intermediate depth between 2 and 5 km. Within this transition layer, both the fracture strength and the frictional strength decrease significantly to levels which could explain the apparent weakness of faults at this depth and deeper, where most of the seismic energy is released, in the depth range between 5 and 15 km. Such a control of pore-pressure on the aftershock decay rate support the increasing body of evidence highlighting the role of fluids along the San Andreas fault system, a necessary ingredient for the weak fault model. Moreover, the present study shows that aftershocks may provide indirect estimates of the magnitude of stress through the variation of well-established statistical parameters. This information could now be used to develop new quantitative methods not only in the physics of faulting and seismic hazard studies but also in industrial settings where deep injection of fluids are known to produce earthquakes.

## Methods

**Frictional rheology**. The Mohr–Coulomb theory proposes that shear failure in a homogeneous medium subject to a uniform triaxial stress $\{\sigma_1, \sigma_2, \sigma_3\}$ should occur

on optimally oriented planes. These planes are parallel to the intermediate shear stress $\sigma_2$ and form an angle $\Theta$ with $\sigma_1$, the maximum compressive stress. Then, the condition for faulting is governed by:

$$\tau \sim \mu \sigma_n, \tag{2}$$

where $\mu$ is a coefficient of static friction, and $\sigma_n$ and $\tau$ are the normal stress and the shear stress resolved on the fault surface. This condition may be rewritten in terms of the principal stresses in order to determine with respect to $\mu$ the angle $\Theta$ and the differential shear stress ($\sigma_1 - \sigma_3$) at which faulting may occur[45]. In this context, the orientation of the principal shear stresses not only determines the faulting mechanism but also the amplitude of the differential shear stress.

The effective overburden stress at a depth $z$ in the crust is:

$$\sigma_v = \rho_r g z (1 - \lambda), \tag{3}$$

where $\rho_r$ is the crustal density, $g$ the acceleration due to gravity, and $0 \leq \lambda \leq 1$ the pore fluid factor. In practice, we have

$$\lambda = \frac{P}{\rho_r g z} = \frac{P}{P_{lith}}, \tag{4}$$

where $P$ is the pore-fluid pressure and $P_{lith} = \rho_r g z$ the lithostatic pressure. Following the Anderson's theory of faulting, the minimum differential stress to initiate slip at a given depth and fluid pressure is:

$$(\sigma_1 - \sigma_3) = \frac{(R' - 1)}{(q(R' - 1) + 1)} \rho_r g z (1 - \lambda), \tag{5}$$

where $q = 0$ for thrust faults,

$$0 < q = \frac{(\sigma_2 - \sigma_3)}{(\sigma_1 - \sigma_3)} < 1 \tag{6}$$

for strike-slip faults, $q = 1$ for normal faults, and

$$R' = \left( \sqrt{1 + \mu^2} - \mu \right)^{-2}. \tag{7}$$

**Pore pressure depth profile**. To explain the vertical $c$-value profiles observed along strike-slip faults in California (Figs. 2, 4), we consider a change in pore pressure regime with respect to depth. In practice, we assume that the pore pressure $P$ operates a transition from hydrostatic to nearly lithostatic as depth exceeds a threshold value $z_c$. Thus, below this critical depth, the reduction of porosity and permeability generate isolated pockets of excess pore pressure. As these pockets increase in number and size according to depth, there is a continuous loss of pore connectivity, which could be associated with a rapid increase in pore pressure. Ultimately, it should be associated with a percolation threshold. Nevertheless, to account for spatial and temporal heterogeneities at the scales of the stacks of aftershocks under examination in this study, we consider an exponential relaxation of the pore fluid factor of the form:

$$\lambda(z) = \begin{cases} \frac{\rho_w}{\rho_r} & \text{for } z < z_c, \\ (1 - \varepsilon) + \left( \frac{\rho_w}{\rho_r} - (1 - \varepsilon) \right) \exp\left( \frac{z_c - z}{L} \right) & \text{for } z \geq z_c, \end{cases} \tag{8}$$

where $\rho_w$ is the density of water, $L$ a characteristic length for a complete loss of pore connectivity, and $\varepsilon$ the ratio of pore pressure to lithostatic stress at great depth (i.e., $z \gg z_c + L$). All the parameter values for this pore pressure depth profile are given in Table 1 (see also the inset in Fig. 4).

**Creep rheology**. Intercrystalline plasticity is a thermally activated process[46], which is responsible for the brittle-ductile transition at depth. For a given strain rate $\dot{\varepsilon}$, the differential shear stress of a steady-state flow of rocks by intercrystalline processes obeys a relationship of the form:

$$(\sigma_1 - \sigma_3) = \left( \frac{\dot{\varepsilon}}{A} \right)^{1/n} \exp\left( \frac{E}{nRT} \right), \tag{9}$$

where $E$ is an activation energy, $n$ the stress exponent, $R$ the gas constant, $T$ the absolute temperature, and $A$ a constant[47].

**Strength envelope**. In order to determine strength envelopes, the strength at any given depth is the lower of the brittle and ductile differential shear stress (see Eqs. (5) and (9), respectively). For the strength envelope in Fig. 4, we consider strike-slip

**Table 1 Model parameter values for the strength envelope shown in Fig. 4**

| Variable | Units | Units |
|---|---|---|
| $g$ | m s$^{-2}$ | 9.81 |
| $\rho_w$ | – | $10^3$ |
| $\rho_r$ | – | $2.3 \times 10^3$ |
| *Friction parameters* | | |
| $\mu$ | – | 0.75 |
| $q$ | – | 0.5 |
| *Creep parameters* | | |
| $\dot{\epsilon}$ | s$^{-2}$ | $10^{-12}$ |
| $A$ | MPa$^{-n}$ s$^{-1}$ | $3.2 \times 10^{-4}$ |
| $n$ | – | 2.30 |
| $R$ | J mol$^{-1}$ K$^{-1}$ | 8.31 |
| $E$ | kJ mol$^{-1}$ | 140 |
| $\partial T / \partial z$ | K m$^{-1}$ | 0.02 |
| *Pore pressure parameters* | | |
| $Z_c$ | m | $2.0 \times 10^3$ |
| $L$ | m | $10^3$ |
| $\varepsilon$ | | 0.1 |

*Note*: In Eq. (9), the temperature $T$ expressed in Kelvin is $T = 273 + \partial T / \partial z$, where $\partial T / \partial z$ is the vertical temperature gradient. See the text for the description of all the other variables. For the friction and creep parameters, all the values come from literature data[47,53]. The parameter $z_c$, $L$, and $\varepsilon$ of the vertical pore pressure profile are the only free parameters used to fit the data

faulting, $\sigma_2 = \sigma_v$ (Supplementary Fig. 1) and

$$\sigma_2 = \frac{\sigma_1 - \sigma_3}{2} \quad \Rightarrow \quad q = \frac{1}{2}. \tag{10}$$

All the other parameter values of Eqs. (5) and (9) are shown in Table 1.

**Space-time parameterisation of the declustering procedure**. Studying more than 30 years of seismicity, the quality of the catalogues may have changed for the magnitude range under investigation. We compare two time periods, 1984–2000 and 2001–2016, considering together all the aftershocks selected along major sub-vertical strike-slip faults in California (Fig. 2). The vertical $c$-value profiles are similar for the two time periods (Supplementary Fig. 2), indicating that there is no influence of catalogue completeness on the depth-dependent behaviour.

Along major strike-slip faults, the majority of slip is confined in narrow layers[48] within a wider zone of damage, the so-called fault zone. To estimate how the vertical $c$-value profile varies across fault zones, we vary the width (Supplementary Fig. 3a) and the position (Supplementary Fig. 3b) of the layer in which we select mainshocks and aftershocks. The depth dependence of the $c$-value is stable up to a distance of 15 km from the faults. At greater distances, $c$-values are larger and show no systematic variation with respect to depth. A tendency of smaller $c$-values at shorter distances from faults is observed in most depth ranges.

The vertical $c$-value profile is computed from the average depth of aftershocks in each depth range. Similar results are obtained using the average depth of mainshocks (Supplementary Fig. 4), except in the neighbourhood of the brittle-creep transition below 13 km, where triggered events are likely to represent a higher proportion of seismicity than in the brittle upper part of the crust[49].

Both the spatial and temporal windows of the declustering procedure scale with the magnitude $M$ of the mainshocks: $R = 0.02 \times 10^{0.5M}$ is the radius of influence expressed in kilometres; $T = 0.04 \times 10^{0.55M}$ is the time of influence expressed in days. Leaving the scaling exponents unchanged, the same vertical $c$-value profiles are obtained when increasing or decreasing $T$ and $R$-values by a factor 2 (Supplementary Fig. 5).

**Influence of the magnitude ranges on the $c$-value depth profile**. All $c$-values are computed for $M_{min}^A \leq M^A \leq M_{max}^A$ aftershocks of $M_{min}^M \leq M^M \leq M_{max}^M$ mainshocks within the time interval $[t_{start}, t_{stop}]$. These six parameters $\{M_{min}^A, M_{max}^A, M_{min}^M, M_{max}^M, t_{start}, t_{stop}\}$ have a regular effect on the $c$-value, which is negligible compared to the influence of depth through the brittle crust. An increase of the $M_A^{min}$-value or a higher range $[M_A^{min}, M_A^{max}]$ of aftershock magnitude lead to smaller $c$-values almost at all depths (Supplementary Fig. 6a, c). Meanwhile, a change in $M_A^{max}$-value for $M_A^{min} = 1.8$ has no influence on $c$-values (Supplementary Fig. 6b). An increase of $M_M^{min}$, $M_M^{max}$ or a higher range $[M_M^{min}, M_M^{max}]$ of mainshock magnitude lead to higher $c$-values, especially close to the depth of the brittle-ductile transition (Supplementary Fig. 6d–f). Finally, $c$-values show no dependence on the time window $[t_{start}, t_{stop}]$ (Supplementary Fig. 6g, h). These variations in $c$-value with the difference between mainshock and aftershock magnitudes are consistent with recent findings from Davidsen and Baiesi[50].

**The declustering method**. Other declustering methods should also lead to same vertical $c$-value profiles. Here, we test an alternative approach dedicated to the identification of earthquake clusters using nearest neighbour distance in time-space-magnitude domains[36–38].

For each pair of earthquakes $\{i, j\}$, we compute the proximity function[51],

$$\nu_{ij} = \begin{cases} t_{ij}(r_{ij})^{d_f} 10^{-bm_i} & \text{for } t_{ij} > 0, \\ +\infty & \text{for } t_{ij} \leq 0. \end{cases} \tag{11}$$

where $t_{ij} = t_j - t_i$ is the event intercurrence time, $r_{ij}$ the spatial distance between the epicentres, $m_i$ the magnitude of event $i$, $d_f$ the fractal dimension of the epicentres, and $b$ the slope of the earthquake-size distribution. Then, we construct families of nearest neighbours using a threshold $\nu_0$ for the proximity function[36,38]. The $\nu_0$-value determines the maximum time-space-magnitude distance between parents and offsprings. Each offspring in a family has only one parent, the closest event using the proximity function. Each parent may have several offsprings. A family may consist of several generations of offsprings. In order to identify direct aftershocks, we select only the first-generation offsprings of the largest magnitude event in each family.

Using the proximity function and first-generation offspring events, Supplementary Figs. 7 and 8 show the same behaviours as in Figs. 2a and 4, respectively. The parameterisation $\{\nu_0, b, d_f\}$ of this new declustering method has no effect on these results for $\nu_0 \in [10^{-7}; 10^{-3}]$, $b \in [0.8; 1.2]$, and $d_f \in [1.3; 2]$ (Supplementary Fig. 9a–c). The similarities between the results obtained by the two declustering methods indicate that changes in $c$-values are not due to artefacts arising from a specific method. In addition, using all generation offsprings, the dependence on depth of the $c$-value is not as clear as the one observed using only first-generation aftershocks (Supplementary Fig. 9d). This confirms the importance of the stress perturbation induced by the mainshock and also suggests that the current parameterisation of our window-based declustering method is likely to isolate aftershock sequences mainly composed of direct aftershocks, i.e., first-generation offsprings (see the $\{M_{min}^A, M_{max}^A, M_{min}^M, M_{max}^M, t_{stop}\}$-values in Supplementary Fig. 6).

**Background activity and $\{c, p\}$-values**. The simultaneous estimation of both $c$ and $p$-values gives the same dependence on depth of the $c$-value as that observed for $p = 1$ (Supplementary Fig. 10). Despite more dispersion of these estimates and a signal with lower amplitude when the $p$-value is not fixed, the same transient increase in $c$-value is observed at intermediate depth between 2 and 5 km. The continuous $c$-value decreases below 6 km exhibit also the same slope in all cases. The $p$-value is quite constant at all depths ($p \lesssim 1$) with variations that show no positive correlation with those of the $c$-value. Then, except for shallow events (<4 km) in southern California, the posterior dependence between $c$ and $p$ has a smaller impact than depth on both parameter estimates.

Except for the first two kilometres, the level of seismic noise in the aftershock sequences is quite uniform at all depths and never exceed 20% (red dots in Supplementary Fig. 10). Over the first day before and after the mainshocks, these levels of noise in the aftershock sequences correspond to seismic rates which are on the same order of magnitude as the rates of foreshock occurrence (dark green dots in Supplementary Fig. 10).

Using our declustering method, many mainshocks are not associated with aftershocks, especially at shallow depth (see pink and light green dots in Supplementary Fig. 10). Considering together all mainshock-aftershocks sequences, our selection procedure agrees with the Bath law that states that a mainshock of $M = 3$ (the mean value of the mainshock magnitude range) should have on average one aftershock with $M \geq 1.8$.

**Northern and southern California**. The depth-dependent behaviour of aftershock statistics are similar for the entire California and separately for northern and southern California (Supplementary Fig. 10). In all areas, a rapid increase in $c$-value at intermediate depth ($\leq 5$ km) is followed by a continuous decrease to depth of about 15 km. The larger number of events in southern California allows to investigate a wider range of depth and reduce uncertainties on both $c$ and $p$-values estimates. In southern California, the $c$-value varies over a range which is an order of magnitude larger than that in northern California. Hence, the Bayesian posterior densities of $\{c, p\}$ in southern California show a more distinct separation between shallow and deep aftershocks than in northern California (Supplementary Fig. 11). In both areas, most of the differences are associated with changes in $c$-values despite the posterior dependency between the parameters $c$ and $p$ of the Modified Omori Law[34].

**A non-parametric estimator of the $c$-value**. Considering a single aftershock sequence, we define $\langle t_g \rangle$ as the geometric mean of elapsed times from mainshocks

to aftershocks within a fixed time window $[t_{start}, t_{stop}]$:

$$\langle t_g \rangle = \sqrt[n]{\prod_{i=1}^{n} t_i} = \exp\left(\frac{1}{n}\sum_{i=1}^{n}\ln(t_i)\right), \qquad (12)$$

where $t_i$, $i \in [1, 2, \ldots, n]$, is the time between the $i$th aftershock in $[t_{start}, t_{stop}]$ and the mainshock. If the aftershock decay rate follows the Omori–Utsu law, $\lambda(t) = K/(t + c)$, Shebalin et al.[35] have shown that $\langle t_g \rangle$ is an implicit function of $c$, $t_{start}$, and $t_{stop}$ (and not $K$). Accordingly, the arithmetic mean of the logarithms of the elapsed times from the mainshock to aftershocks,

$$\ln(\langle t_g \rangle) = \frac{1}{n}\sum_{i=1}^{n}\ln(t_i), \qquad (13)$$

may be used as an estimator for the logarithm of the $c$-value. Because it can be easily compute from aftershock times without a fitting procedure, the $\langle t_g \rangle$-value is first used to identify systematic variations of the characteristic time before the onset of the power-law aftershock decay rate[52]. Then, the $c$-value may be estimated using the solution of an explicit equation (Eq. (9) in Shebalin et al.[35]). This property is valid for a stack of aftershock sequences with a constant $c$-value. For stacks of aftershock sequences with different $c$-values, the distribution will not exactly follow the Omori–Utsu law, but averaging on the logarithm of times still make sense, given an estimate of the corresponding mean value.

Similar dependences on depth are observed using the Bayesian estimates of the $c$-value or the geometric mean $\langle t_g \rangle$ of elapsed times from mainshocks to aftershocks (Supplementary Figs. 12–14). The largest deviations are in cases of small number of aftershocks in the analysis (large error bars). However, the difference between the two procedures may be quite regular in specific seismic zones. For example, the difference is near constant at all depths along the Imperial and Laguna Salada faults. In Geysers and Salton Sea water-injection sites the difference has an opposite sign.

**Data availability**. The data that support the findings of this study are available from the corresponding author upon reasonable request. The python package to perform Bayesian analysis of the Modified Omori Law can be downloaded from http://www.ipgp.fr/~narteau/.

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

## Acknowledgements

We acknowledge financial support of C.N. from the UnivEarthS LabEx programme of Sorbonne Paris Cité (ANR-10-LABX-0023 and ANR-11-IDEX-0005-02), the French National Research Agency (ANR-12-BS05-001-03/EXO-DUNES), and of P.S. from the Russian Science Foundation, Project 16-17-00093.

## Author contributions

P.S. and C.N. designed the study and wrote the manuscript together. P.S. performed the data analysis. C.N. worked on the interpretation of results.

## Additional information

**Competing interests:** The authors declare no competing financial interests.

