## [Peer Review File · Nature Communications]

Reviewers' comments:

Reviewer #1 (Remarks to the Author):

The paper addresses the timely question of triggering processes in seismology and their dependence on stress. Specifically, the paper focuses on aftershock sequences and shows that the characteristic time associated with the triggering rates varies with depth in California. Surprisingly, the variation shows unexpected features and the authors present a simple conceptual model based on different pressure contributions explaining these features. The main new ingredient is a change in the pore pressure variation with depth motivated by porosity changes. This is of direct importance in the context of hydraulic fracturing, a key enabling technology for oil and gas extraction. Given all this, I believe that this paper fulfills the specific criteria of novelty and importance of Nature Communications. I recommend publication provided that all points below are sufficiently addressed.

Major:

1. While the authors show that their findings are robust with respect to variations in the parameters of their methodology, it is not clear whether other methodologies to identify aftershocks lead to the same results. I suggest that the authors test at least one other method, e.g. (i) *Science*, 319, 1076–1079 (2008) or (ii) *J. Geophys. Res. Solid Earth*, 118, 2847–2864 (2013), *J. Geophys. Res. Solid Earth*, 118, 4278–4295 (2013). Both methods would also allow in principle to distinguish between direct (or first generation) aftershocks and indirect ones. The methods used by the authors looks at direct and indirect aftershocks together.

Minor:

1. To address potential issues with catalog completeness for small aftershocks (especially in the Northern California catalog) I suggest to add a figure to the supplemental material which shows the same as Fig.1 in the main paper but separately for Northern California and Southern California.

2. Given the spatial criterion used for identifying main shocks and aftershocks, there should be a boundary effects for depths less than about 1.1km. I suggest that any bias due to this boundary effect is either estimated or the findings for the affected depths are excluded from all figures.

3. The triggering behavior in geothermal areas has been analyzed in these two papers: *Geophysical Research Letters*, 42, 6221-6228 (2015); *Bull. Seismol. Soc. Am.*, 106(3), 846-859 (2016)
They should at least be cited if not discussed.

4. The variations in c with the difference between mainshock magnitude and aftershock magnitude discussed in the supplementary material (Fig. S6) is consistent with other recent findings: *Phys. Rev. E* 94, 022314 (2016)

5. With respect to variations in b -value with depth, there is some more recent work that is important for the discussion here and should be added: *Geophys. Res. Lett.*, 42, 1399–1402 (2015)

6. There are a few crucial typos:

p7: "Above and below this transition layer, the rapid and slow increases in c -value are governed by" should read "decreases".

p8: "between 2 and 5 km. Within this transition layer, both the fracture strength and the frictional strength decrease significantly to levels which could explain the apparent weakness of faults where

most of the seismic energy is released, in the depth range between 5 and 15 km."

The transition layer refers to the 2 to 5km range, so I don't understand the reference to the 5 to

15km range here.

Reviewer #2 (Remarks to the Author):

Assessing the actual stress state of the crust is one of the major problems of seismology and seismic hazard assessment. In this study, the authors use the delay parameter c of the Omori-Utsu aftershock decay law as an estimator of the differential stress variation with depth. They observe for the strike-slip environment in California a significant change of the c -value with depth which can be nicely fitted by the variation of differential stress with a smooth transition from hydrostatic to lithostatic pore-pressure profile in the depth range of 2-5 km and a brittle-ductile transition at around 15 km. The results look very promising and could be, if real, an important result. In the supplementary material, the authors show quite a number of tests to demonstrate the robustness of their results. However, the underlying data are very limited and the results might be biased by some of the underlying assumptions which are not yet tested. These tests are necessary to verify the significance of the results.

(1)

In Figure 1, the authors show that the two different populations of shallow and deep aftershocks have p -values which are significantly smaller than 1. In particular, $p=1$ does not overlap with the corresponding confidence intervals. However, the authors use the constraint $p=1$ for their subsequent analysis. Because the authors provide neither a physical reason/constraint nor observational evidence for $p=1$ (in contrast, Fig.1 shows $p < 1$), this approach is not acceptable. The correct procedure would be to estimate both p and c within the Bayesian approach and then analyze the modal/median/mean value and the confidence interval of the marginal posterior density of c . My guess is that the authors did this in the beginning and recognized that the confidence interval becomes very large. However, to move to an unjustified assumption of $p=1$ because the real uncertainties are too large, is not an option.

(2)

The estimation of the c -value is a sensitive parameter which might be affected by incompleteness (which is avoided in this study by analyzing only small mainshocks) and background contamination. Background activity which occurs by chance (independently of the mainshock) within the spatiotemporal selection windows will not follow the Omori-Utsu decay and thus will blur the results. For example, if the Omori-Utsu law would be fitted to a constant activity of background events, the fit would lead to an estimated c -value which is longer than the fitted time period in order to simulate the constant rate. In general, the c -value is expected to increase if the aftershock activity is contaminated by independent events. Thus it is important to test whether background activity can have an effect on the presented results. It is particularly suspicious that the maximum c -value shown in Fig.3 for the individual fault segments are observed in the depth range where most earthquakes (background activity) occur (5-7 km). Furthermore, based on the Bath law a mainshock of $M=3$ (the mean value of the mainshock interval used by the authors) should have on average one aftershock with $M \geq 1.8$. However, the authors found for 3000 mainshocks more than 4800 aftershocks which might indicate a significant contamination of the data by independent events.

In order to test this potentially critical issue, I suggest to do the following:

- A. Provide a histogram of the depth distribution of the mainshocks, aftershocks and background events. This could be also used to see whether a simple correlation between the $\log(c)$ -value and N exists.
- B. Count the average number of mainshock-independent events expected in the spatiotemporal selection window based on the number of events occurring in the same spatiotemporal intervals

preceding the mainshocks. In this way, it can be evaluated whether or not the contamination could bias the c-value estimations.

C. If B shows a significant contribution, the c-value estimation should be repeated for the fit of the Omori-Utsu decay PLUS a background term.

(3)

The authors state that "despite some variations related to the spatial distribution of seismicity in these specific areas, the same behavior may be observed across the entire California: a sharp increase in c-value at shallow depth (< 5 km) is followed by a continuous decrease to depth of about 15 km".

Looking at Fig.2, I can recognize a sharp increase for < 5km only for 4 out of 12 cases, namely Imperial, Calaveras, San Andreas creeping section and Maacama, while in 8 out of 12 cases, the c-value is constant (within its error bounds). Furthermore, also another feature of the curve for the cumulative activity of all regions shown in Fig. 4 does not show up in the majority of fault segments: The sharp decrease of the c-value at very shallow depth (< 2 km) is seen mainly in Southern San Andreas but not in other regions (e.g. Maacama). This raises the question whether the cumulative curve does represent any meaningful result or it is an almost random superposition of very different local properties.

(4)

The current plots pretend to have a much higher resolution in depth (0.3 km step size) than they actually have, because of the relative large bin width of 3 km. Thus the results for non-overlapping depth bins should be also presented (maybe even in the same figures).

Other points:

- my suggestion for an alternative title: "Depth dependent stress revealed by aftershocks"

- Last paragraph:

The statement "using the myriad of small magnitude events" is not in agreement with the actually very limited number of currently available events to perform such as study.

The limited number of events also restricts the possibility (which is mentioned in the last sentences of the article) to resolve details of the spatiotemporal fluctuations of the stress field.

- captions of Fig. 2 & 4: "central value of the 3 km depth bin for selecting aftershocks" instead of "mean depth of the selected aftershocks"

- Supplement, Eq.(7): A minus sign is missed in the argument of the exponential function.

- Supplement, below Eq.(9): " $K / (t + c)$ " instead of " $K / (1 + c)$ "

Reviewer #3 (Remarks to the Author):

The authors make very interesting physical claims and hypotheses based on the statistical variation of estimates of the c parameter with depth. The idea of subdividing catalogs in this way and comparing how estimates vary with depth is a really good one, and the paper is largely well written and sensible. I have some comments and questions for the authors though.

1. Declustering is always a questionable procedure, as it seems to be inherently dependent on rather ambiguous choices and results in loss of information. Here the choice of method for declustering seems quite reasonable, and I have no reason to suspect that the authors tried multiple different possibilities and are only reporting certain results. Nevertheless, I do question why declustering was needed here and what the advantage is of doing it at all. If declustering

must be done, a good alternative to that done here is the stochastic declustering method of Zhuang et al. (2002). It would be interesting for the authors to comment on whether the results would be different had this stochastic declustering been performed.

2. On p4, the authors note that their choice of constraints on the catalog reduce artifacts related to catalog completeness. This is really important, especially when considering the c-value. On the other hand, I am not sure I agree that the aftershocks are sufficiently large here to guarantee completeness. I doubt these catalogs are really complete down to M1.8. No way, especially at depth.

3. On p4 in the middle, by "Bayesian statistics", the authors really mean and should say "Bayesian estimates of the parameters", and specify that uniform priors were used, or whatever priors were chosen.

4. The work of Kagan seems to be glaringly missing from the references, especially when discussing critical things like the dependency between c and p values, and the variations in parameters spatially as in Kagan et al. (2010) or Chu et al. (2011).

5. On the top of p5, in discussing fig2 I wouldn't overstate the similarity of these curves too much, especially given the relatively large standard errors and the fact that there is some disagreement from location to location. I think the writing is a little too strong here.

6. All of these earthquakes are quite shallow. The depths here appear to be limited to just 15 or occasionally 20km. Why not consider deeper earthquakes too?

Chu, A., Schoenberg, F.P., Bird, P., Jackson, D.D., and Kagan, Y.Y. (2011). Comparison of ETAS parameter estimates across different global tectonic zones. *BSSA*, 101(5), 2323-2339.

Kagan, Y.Y. (2002). Modern California earthquake catalogs and their comparison, *Seismological Research Letters*, 73(6), 921-929.

Kagan, Y.Y., P. Bird, and D.D. Jackson (2010). Earthquake patterns in diverse tectonic zones of the globe, *Pure Appl. Geoph.*, (The Frank Evison Volume), 545 167(6/7).

Zhuang, J., Y. Ogata, and D. Vere-Jones (2002). Stochastic declustering of space-time earthquake occurrences. *Journal of the American Statistical Association*, 97(458), 369-380.

ANSWERS TO REVIEWER 1 (R1)

The paper addresses the timely question of triggering processes in seismology and their dependence on stress. Specifically, the paper focuses on aftershock sequences and shows that the characteristic time associated with the triggering rates varies with depth in California. Surprisingly, the variation shows unexpected features and the authors present a simple conceptual model based on different pressure contributions explaining these features. The main new ingredient is a change in the pore pressure variation with depth motivated by porosity changes. This is of direct importance in the context of hydraulic fracturing, a key enabling technology for oil and gas extraction. Given all this, I believe that this paper fulfills the specific criteria of novelty and importance of Nature Communications. I recommend publication provided that all points below are sufficiently addressed.

Major

While the authors show that their findings are robust with respect to variations in the parameters of their methodology, it is not clear whether other methodologies to identify aftershocks lead to the same results. I suggest that the authors test at least one other method, e.g. (i) Science, 319, 10761079 (2008) or (ii) J. Geophys. Res. Solid Earth, 118, 28472864 (2013), J. Geophys. Res. Solid Earth, 118, 42784295 (2013). Both methods would also allow in principle to distinguish between direct (or first generation) aftershocks and indirect ones. The methods used by the authors looks at direct and indirect aftershocks together.

In this new version of the manuscript, we have also used the declustering method proposed by *Zaliapin and Ben-Zion (2013)*, which is based on the identification of nearest-neighbours in time-space-magnitude domains (see also *Baiesi & Paczuski (2004)*, *Gu et al. (2013)* and *Zaliapin & Ben-Zion (2016)*). The new Supplementary Figs. 7, 8 and 9 can be compared to Figs. 2 and 4 of the main manuscript. The results obtained with both declustering methods are consistent with each other, indicating that the dependence of the c -value on depth is not strongly affected by the details of the declustering method. Because we use only first generation aftershock in the new declustering procedure, these results suggest that our method is likely to contain an important proportion of direct aftershocks.

A new paragraph is dedicated to this study in the new ‘Methods’ section (see also Supplementary Figs. 7, 8 and 9).

Minors

1. To address potential issues with catalog completeness for small aftershocks (especially in the northern California catalog) I suggest to add a figure to the supplemental material which shows the same as Fig. 1 in the main paper but separately for northern California and southern California.

This new figure has been added to the Supplementary Information which is discussed in the new ‘Methods’ section in a paragraph dedicated to the estimation of both p and c -values (Supplementary Fig. 11). Note that, according to a remark of R2, we have changed Fig. 1 to show that in a vast majority of cases the dispersion of p -value estimates extends also over the range $p > 1$.

2. Given the spatial criterion used for identifying main shocks and aftershocks, there should be a boundary effects for depths less than about 1.1 km. I suggest that any bias due to this boundary effect is either estimated or the findings for the affected depths are excluded from all figures.

Using both types of declustering methods, there is almost no selected aftershocks at depth smaller than 1 km (Fig.). In addition, we do not study stacked aftershock sequences in which the number of events is smaller than 20. The c -values are shown at the mean depth of the events in the stacked aftershock sequences. All these conditions together suggest that there is no bias due to a top boundary effect.

3. The triggering behavior in geothermal areas has been analyzed in these two papers: Geophysical Research Letters, 42, 6221-6228 (2015); Bull. Seismol. Soc. Am., 106(3), 846-859 (2016) They should at least be cited if not discussed.

We have cited these two papers where we introduce triggered seismicity in geothermal areas.

4. The variations in c with the difference between mainshock magnitude and aftershock magnitude discussed in the supplementary material (Fig. S6) is consistent with other recent findings: Phys. Rev. E 94, 022314 (2016)

We have added this remark and this reference to the new ‘Methods’ section.

5. With respect to variations in b -value with depth, there is some more recent work that is important for the discussion here and should be added: Geophys. Res. Lett., 42, 13991402 (2015)

We have added this citation where we discuss the stress dependence of the earthquake b -value.

6. There are a few crucial typos:

p7: "Above and below this transition layer, the rapid and slow increases in c -value are governed by" should read "decreases".

Done

p8: "between 2 and 5 km. Within this transition layer, both the fracture strength and the frictional strength decrease significantly to levels which could explain the apparent weakness of faults where most of the seismic energy is released, in the depth range between 5 and 15 km." The transition layer refers to the 2 to 5 km range, so I don't understand the reference to the 5 to 15 km range here.

We have rephrased this sentence to be more clear. Basically, we have an abrupt drop in strength between 2 to 5 km which is only slowly compensated by the lithostatic stress at higher depths given the apparent coefficient of friction.

ANSWERS TO REVIEWER 2 (R2)

Assessing the actual stress state of the crust is one of the major problems of seismology and seismic hazard assessment. In this study, the authors use the delay parameter c of the Omori-Utsu aftershock decay law as an estimator of the differential stress variation with depth. They observe for the strike-slip environment in California a significant change of the c -value with depth which can be nicely fitted by the variation of differential stress with a smooth transition from hydrostatic to lithostatic pore-pressure profile in the depth range of 2-5 km and a brittle-ductile transition at around 15 km. The results look very promising and could be, if real, an important result. In the supplementary material, the authors show quite a number of tests to demonstrate the robustness of their results. However, the underlying data are very limited and the results might be biased by some of the underlying assumptions which are not yet tested. These tests are necessary to verify the significance of the results.

We would like to thank R2 for this complete and sharp review that gives an overview of all the statistical procedures that we have implemented and tested over the last few years studying aftershocks.

To answer to most of the comments of R2 keeping a reasonable size for the new ‘Methods’ section we have chosen to show the depth dependence of both p and c -values for two declustering methods estimating simultaneously the background seismicity in southern and northern California. In what follows, we sequentially answer to all of the remarks of R2 to reinforce the significance and the stability of the results.

(1) In Figure 1, the authors show that the two different populations of shallow and deep aftershocks have p -values which are significantly smaller than 1. In particular, $p = 1$ does not overlap with the corresponding confidence intervals. However, the authors use the constraint $p = 1$ for their subsequent analysis. Because the authors provide neither a physical reason/constraint nor observational evidence for $p = 1$ (in contrast, Fig. 1 shows $p < 1$), this approach is not acceptable. The correct procedure would be to estimate both p and c within the Bayesian approach and then analyze the modal/median/mean value and the confidence interval of the marginal posterior density of c . My guess is that the authors did this in the beginning and recognized that the confidence interval becomes very large. However, to move to an unjustified assumption of $p = 1$ because the real uncertainties are too large, is not an option.

We consider that the best option in exploring the time delay before the onset of the power-law decay rate is to compute the non-parametric estimator $\langle t_g \rangle$ (Shebalin *et al.*, 2011), the geometric mean of aftershock times over a given time period $[t_{\text{start}}, t_{\text{stop}}]$ (see the new ‘Methods section’).

In this case, there is no artefact related to the fitting procedure and the codependence between the model parameters (*Holschneider et al., 2012*). In addition, there are direct relations between $\langle t_g \rangle$ and c -values for fixed p -values (*Shebalin et al., 2011*). We also want to take advantage of the well-known Modified Omori Law and make our work more accessible to a wider community. Then, we systematically fix the p -value to 1 to study variations of c -value. Thus, we know that we will recover with the c -value the signal observed with the $\langle t_g \rangle$ -value. In this new version of the manuscript, we have kept the same approach and still show in the main manuscript the result obtained with $p = 1$. Note that similar results could have been obtained with other p -values, for example with $p < 1$ as it is commonly observed in California (Supplementary Fig. 10).

Meanwhile, we fully agree with the comment of R2 and we have to demonstrate that the depth-dependent behaviour of the c -value is not only due to continuous changes in p -value. In Supplementary Fig. 10, we simultaneously estimate the c and p -values and compare these estimates with results obtained with $p = 1$. This is discussed in a paragraph of the new ‘Methods’ section.

(2) The estimation of the c -value is a sensitive parameter which might be affected by incompleteness (which is avoided in this study by analyzing only small mainshocks) and background contamination. Background activity which occurs by chance (independently of the mainshock) within the spatiotemporal selection windows will not follow the Omori-Utsu decay and thus will blur the results. For example, if the Omori-Utsu law would be fitted to a constant activity of background events, the fit would lead to an estimated c -value which is longer than the fitted time period in order to simulate the constant rate. In general, the c -value is expected to increase if the aftershock activity is contaminated by independent events. Thus it is important to test whether background activity can have an effect on the presented results. It is particularly suspicious that the maximum c -value shown in Fig. 2 for the individual fault segments are observed in the depth range where most earthquakes (background activity) occur (5-7 km). Furthermore, based on the Bath law a mainshock of $M = 3$ (the mean value of the mainshock interval used by the authors) should have on average one aftershock with $M \geq 1.8$. However, the authors found for 3000 mainshocks more than 4800 aftershocks which might indicate a significant contamination of the data by independent events.

We only refer here to mainshocks with aftershocks. As shown in Supplementary Fig. 10 (pink and light green dots), many mainshocks have no aftershock, especially at shallow depth. All together, the numbers of selected mainshocks and aftershocks agree with the Bath’s law.

In order to test this potentially critical issue, I suggest to do the following:

- A. Provide a histogram of the depth distribution of the mainshocks, aftershocks and background events. This could be also used to see whether a simple correlation between the $\log(c)$ -value and N exists.

In addition to the simultaneous estimation of c and p -values, we have also provided the depth distribution of mainshocks and aftershocks (see Supplementary Fig. 10). Using the Modified Omori Law, *Holschneider et al. (2012)* have shown that the productivity is decoupled from the other parameters c and p . The dependence of the c -value on the differential shear stress and the transition from hydrostatic to near-lithostatic pore-pressure conditions explain why highest c -values are observed in the depth range where most earthquakes occur.

- B. Count the average number of mainshock-independent events expected in the spatiotemporal selection window based on the number of events occurring in the same spatiotemporal intervals preceding the mainshocks. In this way, it can be evaluated whether or not the contamination could bias the c -value estimations.

We have compared the seismic rate over one day before the mainshock to the aftershock rate over the same spatiotemporal intervals (dark green dots in Supplementary Fig. 10).

- C. If B shows a significant contribution, the c -value estimation should be repeated for the fit of the Omori-Utsu decay PLUS a background term.

The background term has been added to estimations of c with $p = 1$ (white dots in Supplementary Fig. 10) and to the simultaneous estimations of c and p -values (black and blue dots in Supplementary Fig. 10). The noise level never exceed 20% except at shallow depth in southern California where the codependence between the c and p -value estimates is also observed. This level of noise do not affect the depth-dependent behaviour of the c -value.

All these results are discussed in the new ‘Methods’ section and summarised in the main text.

(3) The authors state that ”despite some variations related to the spatial distribution of seismicity in these specific areas, the same behavior may be observed across the entire California: a sharp increase in c -value at shallow depth (< 5 km) is followed by a continuous decrease to depth of about 15 km”. Looking at Fig. 2, I can recognize a sharp increase for < 5 km only for 4 out of 12 cases, namely Imperial, Calaveras, San Andreas creeping section and Maacama, while in 8 out of 12 cases, the c -value is constant (within its error bounds). Furthermore, also another feature of the curve for the cumulative activity of all regions shown in Fig. 4 does not show

up in the majority of fault segments: The sharp decrease of the c -value at very shallow depth (< 2 km) is seen mainly in southern San Andreas but not in other regions (e.g. Maacama). This raises the question whether the cumulative curve does represent any meaningful result or it is an almost random superposition of very different local properties.

This is correct that the writing concerning the similarities is too strong (see also a remark of R3). We have modified this paragraph to emphasise that local properties along faults in California differ and that we rely on a limited amount of data. However, we continue to consider that the overall effect of the confining stress and pore-pressure can be averaged at the length scale of California considering only events along major strike-slip faults. The mapping issue is a different problem, which will be addressed later.

(4) The current plots pretend to have a much higher resolution in depth (0.3 km step size) than they actually have, because of the relative large bin width of 3 km. Thus the results for non-overlapping depth bins should be also presented (maybe even in the same figures).

We have specified in the main text that in the step that

“In all places, the continuous change in c -value should be considered with respect to the depth interval of the overlapping sliding-window for event selection.”

Other points:

- my suggestion for an alternative title: "Depth dependent stress revealed by aftershocks". We have changed the title. Thank you for this suggestion.
- Last paragraph: The statement "using the myriad of small magnitude events" is not in agreement with the actually very limited number of currently available events to perform such as study. The limited number of events also restricts the possibility (which is mentioned in the last sentences of the article) to resolve details of the spatiotemporal fluctuations of the stress field.
We have removed the adjective "*myriad*".
- captions of Fig. 2 & 4: "central value of the 3 km depth bin for selecting aftershocks" instead of "mean depth of the selected aftershocks".
In Figs. 2, 3 and 4, we have specified "*... the mean depth of the selected aftershocks within the 3 km depth bins ...*".
- Supplement, Eq.(7): A minus sign is missed in the argument of the exponential function.
Done.

- Supplement, below Eq.(9): " $K / (t + c)$ " instead of " $K / (1 + c)$ "
Done.

ANSWERS TO REVIEWER 3 (R3)

The authors make very interesting physical claims and hypotheses based on the statistical variation of estimates of the c parameter with depth. The idea of subdividing catalogs in this way and comparing how estimates vary with depth is a really good one, and the paper is largely well written and sensible. I have some comments and questions for the authors though.

1. Declustering is always a questionable procedure, as it seems to be inherently dependent on rather ambiguous choices and results in loss of information. Here the choice of method for declustering seems quite reasonable, and I have no reason to suspect that the authors tried multiple different possibilities and are only reporting certain results. Nevertheless, I do question why declustering was needed here and what the advantage is of doing it at all. If declustering must be done, a good alternative to that done here is the stochastic declustering method of Zhuang et al. (2002). It would be interesting for the authors to comment on whether the results would be different had this stochastic declustering been performed.

We agree that declustering is always a questionable procedure. We need it here to avoid analysing secondary aftershocks and foreshocks. Indeed, the dependency of the c -value on stress relies on the amplitude of the stress perturbation induced by the mainshocks, and it is impossible to estimate such a perturbation for a cascade of aftershocks.

In this new version of the manuscript, as said in the answer to R1, we have also used the declustering method proposed by *Zaliapin and Ben-Zion (2013)*, which is based on the identification of nearest-neighbours in time-space-magnitude domains (see also *Baiesi & Paczuski (2004)*, *Gu et al. (2013)* and *Zaliapin & Ben-Zion (2016)*). The new Supplementary Figs. 7, 8 and 9 can be compared to Figs. 2 and 4 of the main manuscript. The results obtained with both declustering methods are consistent with each other, indicating that the dependence of the c -value on depth is not strongly affected by the details of the declustering method. Because we use only first generation aftershocks in the new declustering procedure, these results also suggest that our method is likely to contain an important proportion of direct aftershocks.

A new paragraph is dedicated to this study in the new ‘Methods’ section (see also Supplementary Figs. 7, 8 and 9).

2. On p4, the authors note that their choice of constraints on the catalog reduce artifacts related to catalog completeness. This is really important, especially when considering the c -value. On the other hand, I am not sure I agree that the aftershocks are sufficiently large here to guarantee completeness. I doubt these catalogs are really complete down to M1.8. No way, especially at depth.

For the magnitude of completeness, we rely on the study of *Schorlemmer and Woessner (2008)*. Considering only events along strike-slip fault in California, $M_c = 1.8$ appears as a conservative value. Along the San Andreas and the Hayward fault our own computation gives also $M_c \leq 1.8$ (see Supplementary Information of *Vorobieva et al., Geoph. Res. Lett., 43, 6869-6875, 2016*). Most importantly, as shown in Supplementary Fig. 6, the aftershock magnitude range has no impact on the depth-dependent behaviour of the c -value, indicating that completeness is not a major issue.

3. On p4 in the middle, by "Bayesian statistics", the authors really mean and should say "Bayesian estimates of the parameters", and specify that uniform priors were used, or whatever priors were chosen.

Done.

4. The work of Kagan seems to be glaringly missing from the references, especially when discussing critical things like the dependency between c and p values, and the variations in parameters spatially as in Kagan et al. (2010) or Chu et al. (2011).

We have cited these two papers in the second paragraph of the main text.

5. On the top of p5, in discussing fig2 I wouldn't overstate the similarity of these curves too much, especially given the relatively large standard errors and the fact that there is some disagreement from location to location. I think the writing is a little too strong here.

We have modified this paragraph to emphasise that local properties along faults in California differ and that we rely on a limited amount of data.

6. All of these earthquakes are quite shallow. The depths here appear to be limited to just 15 or occasionally 20 km. Why not consider deeper earthquakes too?

There is simply no events deeper.

REVIEWERS' COMMENTS:

Reviewer #1 (Remarks to the Author):

The authors have convincingly addressed all points I raised in my previous report. I also feel that their reply to the other comments and remarks is satisfactorily. Thus, I recommend publication of the paper in its current form.

Reviewer #2 (Remarks to the Author):

The authors have met all my points and present now additional material in the supplementary material which helps to clarify the robustness of the results. The signal of the depth-dependent c-value variation is quite weak, as e.g. shown by the overlapping confidence intervals in the case of the simultaneous estimation of c & p in supplementary Fig.10. However, the trend of the results are found to be similar, independent of the cluster selection method and background activity. Thus I recommend the publication of these results.

I have only two remaining comments:

1. The analysis of clusters selected with the nearest-neighbor method only shows a similar clear c-value variation (compared to their original method) for direct aftershocks, while the result for all aftershocks is almost constant with depth. The authors explain this observation by stating that "our declustering method is likely to isolate aftershock sequences mainly composed of direct aftershocks". However, their method is a window-based method which selects by definition all generations of aftershocks simultaneously and is not selective for direct aftershocks. Therefore, the authors should rethink their explanation.
2. With regard to my 4th point of the first review, which concerns the heavily overlapping depth bins, the authors introduced a new sentence: "In all places, the continuous change in c-value should be considered with respect to the depth interval of the overlapping sliding-window for event selection". This sentence is not clear and should be reformulated.

Reviewer #3 (Remarks to the Author):

The authors addressed my concerns admirably and I am now very satisfied with accepting the manuscript.

ANSWERS TO REVIEWER 2 (R2)

The authors have met all my points and present now additional material in the supplementary material which helps to clarify the robustness of the results. The signal of the depth-dependent c -value variation is quite weak, as e.g. shown by the overlapping confidence intervals in the case of the simultaneous estimation of c & p in supplementary Fig. 10. However, the trend of the results are found to be similar, independent of the cluster selection method and background activity. Thus I recommend the publication of these results.

I have only two remaining comments:

1. The analysis of clusters selected with the nearest-neighbor method only shows a similar clear c -value variation (compared to their original method) for direct aftershocks, while the result for all aftershocks is almost constant with depth. The authors explain this observation by stating that "our declustering method is likely to isolate aftershock sequences mainly composed of direct aftershocks". However, their method is a window-based method which selects by definition all generations of aftershocks simultaneously and is not selective for direct aftershocks. Therefore, the authors should rethink their explanation.

We have specified that this is the parameterization of our declustering method that allows to obtain a high proportion of direct aftershocks in the stacks of aftershocks.

2. With regard to my 4th point of the first review, which concerns the heavily overlapping depth bins, the authors introduced a new sentence: "In all places, the continuous change in c -value should be considered with respect to the depth interval of the overlapping sliding-window for event selection". This sentence is not clear and should be reformulated.

We have changed this sentence to explain how the sliding window algorithm gradually replaces individual events in the stacks of aftershocks.